# Structure-Activity Relationship Investigations of Novel Constrained Chimeric Peptidomimetics of SOCS3 Protein Targeting JAK2

**DOI:** 10.3390/ph15040458

**Published:** 2022-04-09

**Authors:** Sara La Manna, Marilisa Leone, Flavia Anna Mercurio, Daniele Florio, Daniela Marasco

**Affiliations:** 1Department of Pharmacy, Research Center on Bioactive Peptides (CIRPEB), University of Naples “Federico II”, 80131 Naples, Italy; sara.lamanna@unina.it (S.L.M.); daniele.florio@unina.it (D.F.); 2Institute of Biostructures and Bioimaging (CNR), 80145 Naples, Italy; marilisa.leone@cnr.it (M.L.); flaviaanna.mercurio@cnr.it (F.A.M.)

**Keywords:** mimetic peptides, cytokine signaling, JAK/STAT, SOCS3, stapled peptides

## Abstract

SOCS3 (suppressor of cytokine signaling 3) protein suppresses cytokine-induced inflammation and its deletion in neurons or immune cells increases the pathological growth of blood vessels. Recently, we designed several SOCS3 peptidomimetics by assuming as template structures the interfacing regions of the ternary complex constituted by SOCS3, JAK2 (Janus Kinase 2) and gp130 (glycoprotein 130) proteins. A chimeric peptide named KIRCONG chim, including non-contiguous regions demonstrated able to bind to JAK2 and anti-inflammatory and antioxidant properties in VSMCs (vascular smooth muscle cells). With the aim to improve drug-like features of KIRCONG, herein we reported novel cyclic analogues bearing different linkages. In detail, in two of them hydrocarbon cycles of different lengths were inserted at positions i/i+5 and i/i+7 to improve helical conformations of mimetics. Structural features of cyclic compounds were investigated by CD (Circular Dichroism) and NMR (Nuclear Magnetic Resonance) spectroscopies while their ability to bind to catalytic domain of JAK2 was assessed through MST (MicroScale Thermophoresis) assay as well as their stability in biological serum. Overall data indicate a crucial role exerted by the length and the position of the cycle within the chimeric structure and could pave the way to the miniaturization of SOCS3 protein for therapeutic aims.

## 1. Introduction

Suppressor of cytokine signaling (SOCSs) are a family of cytokine-inducible proteins able to inhibit cytokine signaling mainly through the negative regulation of the Janus kinase/signal transducer and activator of transcription (JAK/STAT) pathway [1]. JAK/STAT is a major contributor of chronic inflammatory diseases and is largely involved in the regulation of the expression of many genes involved in cellular activation, differentiation, migration, apoptosis, and proliferation [2].

This protein family includes eight members: SOCS1-SOCS7 and cytokine-inducible SH2-containing protein (CIS). Structurally, these proteins contain a Src homology 2 (SH2) domain, a variable N-terminal domain and a C-terminal SOCS box [3]. Two members of this family, SOCS1 and SOCS3 are the unique to have a motif in the N-terminal region, called kinase inhibitory region (KIR) [4] that is crucially involved in the inhibition of JAKs [5]. Biochemical studies have highlighted different mechanisms of action (MOA) of these proteins in the inhibition of JAKs: while the SH2 domain of SOCS1 directly binds to the activation loop of JAK [6], the SH2 domains of CIS, SOCS2 and SOCS3 bind to phosphorylated tyrosine residues on activated cytokine receptors (glycoprotein (gp) 130 in IL-6 signaling) [7]. Dysfunctions of and genetic alterations in JAK/STAT/SOCS axis are linked with a wide range of cardiovascular [8], inflammatory and autoimmune disorders [9]. SOCS1 is a critical regulators of IFNs signaling [10,11] and interleukins (IL) as −2 [12], −12, −23 [13], −6 [14]. Its deficiency leads to inflammation [15], while its overexpression represses pro-inflammatory genes [16] and apoptosis caused by reactive oxygen species (ROS) [17,18]. In atherosclerotic models, the inhibition of JAK2, STAT1 and STAT3 prevented lesion formation [19,20]. Otherwise, mice lacking SOCS1 (Socs1^−/−^) resulted protected from viral infection [21,22]. 

SOCS3 has a key role in controlling IL-6 signaling in the hepatocyte priming stage and it attenuates hepatocyte proliferation [23]. SOCS3 overexpression revealed favorable effects in colorectal [24], ovarian cancer lines [25] in MCF7 BC (Breast Cancer) cells and several solid tumors [26]. It represses cell proliferation through the reduction of STAT3 expression [27] and tumor growth and metastasis formation in mouse xenograft models [28]. In non-small cell lung cancers (NSCLC) it inhibits many tumor cell functions [29,30]; while a downregulation of SOCS3 was found in gastric adenocarcinoma tissues [31] and linked with enhanced risks of recurrent disease in BC patients [32]. In the chronic constriction injury of the sciatic nerve (CCI), SOCS3 prevented the abnormal expression of IL-6 and C–C motif chemokine ligand 2 (CCL2) and attenuated allodynia in rats [33]. A direct link with cardiovascular diseases was often observed: in diabetic cardiac fibrosis tissue, diabetic cardiomyopathy (DCM) patients’ heart tissue and cardiac fibroblasts (after long term high-glucose treatment), SOCS3 resulted downregulated while its overexpression inhibited cardiac fibroblast activation and collagen production [34]. 

Interestingly, in neovascular age-related macular degeneration (nAMD), SOCS3 overexpression in myeloid lineage cells suppressed laser-induced choroidal neovascularization (CNV) through the block of myeloid lineage-derived macrophage/microglia recruitment and proinflammatory factors [35].

In this *scenario*, the identification of SOCS 1, 3 proteomimetics endowed with anti-inflammatory/antioxidant properties, is considered a valid strategy to employ peptide-based compounds as novel therapeutics [36]. For SOCS1, the miniaturization process is well advanced since our [37,38,39,40] and other research groups [41,42,43,44,45,46,47], starting from KIR domain (52–67), developed promising peptidomimetics with important anti-inflammatory properties. Very recently, in turn, several JAK1 peptidomimetics able to interact tightly with the SOCS1-SH2 domain and to block its activity have been designed as potential antiviral drugs [48].

Conversely, given the structural complexity of the ternary complex formed by SOCS3 with JAK2 and gp130 [7], reports on SOCS3 mimetics have been developed only from our research activity, at the best of our knowledge. Few years ago, we identified, a linear peptide spanning 22–45 residues of SOCS3, including KIR and ESS (Extended SH2 Subdomain) regions, called KIRESS. Intratumoral administration of this peptide significantly reduced growth of squamous cell carcinoma [49] and in triple-negative breast cancer (TNBC) it prevented the formation of pulmonary metastasis [50] and it was applied as an in vivo mimetic of the whole SOCS3 protein [35].

Recently, we designed another SOCS3 mimetic, named KIRCONG chim, that includes non-contiguous protein regions. It demonstrated able to bind to the catalytic domain of JAK2 and to act as a potent anti-inflammatory and antioxidant agent in VSMCs (vascular smooth muscle cells) [51].

Herein, to improve the stability to proteolytic degradation and to rigidify the structure, we introduced conformational constraints into the linear KIRCONG chim sequence: in detail, four cyclic compounds were designed and their conformational features analyzed through CD (Circular Dichroism) and NMR (Nuclear Magnetic Resonance) spectroscopies while their binding abilities were assessed by MST (MicroScale Thermophoresis) technique. To evaluate if introduced chemical modifications could provide major stability with respect to linear peptide in cellular contexts, serum stability assays were carried out.

## 2. Materials and Methods

### 2.1. Peptide Synthesis

Peptides were synthesized as carboxyl C-termini on Wang resin, using 9-fluorenylmethoxycarbonyl/*tert*-butyl (Fmoc/*t*Bu) strategy. For KIRCONG *amide*, the formation of a lactam bridge was obtained on solid support, by employing the super-acid labile protecting groups of side chains in Fmoc-Lys(Mtt)-OH and Fmoc-Glu(O-2-PhiPr)OH derivatives [37]. For KIRCONG *disulfide*, the peptide was dissolved in sodium carbonate 100 mM (0.1 mg/mL) and the mixture was left open to atmosphere under magnetic stirring, until the intramolecular oxidation was complete (confirmed by LC-MS analysis). Stapled analogues, KIRCONG *i/i+5* and i/i+7, were obtained via ruthenium-based ring-closing metathesis (RCM) of olefin-derivatized amino acid residues (2-(4′-pentenyl)alanine and 2-(7′-octenyl)alanine) at the (i) and (i+5) or (i+7) positions in the peptide backbone. Peptides were purified through RP-HPLC and identified as already reported [52]. Purified peptides were lyophilized and stored at −20 °C until use.

### 2.2. Shake Flask Procedure for Determination of Log P

The logarithmic partition coefficient Log P between 1-octanol and water phases was determined for KIRCONG analogues using the shake flask method [53]. All peptides were dissolved in an equal volume of water (pre-saturated with 1-octanol) and 1-octanol (pre-saturated with ultrapure water) to reach a final concentration of each peptide of 300 µM. The mixtures were shaken mechanically for 120 min at 25 °C. The samples were centrifuged to assist with bilayer formation. The experiments were performed at least in duplicates.

The concentrations in water phase were determined by UV/Vis absorption (BioDrop-DUO-spectrophotometer, Biochrom, Waterbeach Cambridge, UK) employing as ε_275 nm_ = 8450 M^−1^ cm^−1^, due to the presence of two Phe, two Tyr and one Trp residues in the sequences. The Log P was calculated according to the following equation, by assuming C_octanol_ = C_total_ − C_water_ (C: peptide concentration):$$\log P=\log\frac{C_{octanol}}{C_{water}\;}$$

### 2.3. Circular Dichroism (CD) Spectroscopy

CD spectroscopy experiments were carried out by employing a Jasco J-815 spectropolarimeter (JASCO, Tokyo, Japan), at room temperature in the spectral range 190–260 nm and spectra are averaged over two scans, to which blanks were subtracted. CD signals were converted to mean residue ellipticity with deg* cm^2^*dmol^−1^*res^−1^ as units. Scan speed value was 20 nm/min, band width 2.0 nm, resolution 0.2 nm, sensitivity 50 mdeg and response 4 s. Samples were prepared by dilution of freshly prepared stock solutions in 100% TFE (2,2,2-Trifluoroethanol) (1 mM on average). In the CD samples compound concentrations were 100 μM. Spectra were acquired in a quartz cuvette with a path-length of 0.1 cm in mixtures TFE/phosphate buffer, 10 mM at pH 7.4 [52].

### 2.4. NMR Studies 

NMR spectra of KIRCONG i/i+5 and KIRCONG i/i+7 stapled peptides were registered at 25 °C on a Varian Unity Inova 600 MHz spectrometer provided with a cold probe. KIRCONG i/i+5 NMR experiments were acquired in H_2_O/TFE (2,2,2-trifluoroethanol-D3 −99.5% isotopic purity, Sigma-Aldrich, Milan, Italy) 60/40 *v*/*v* and 85/15 *v*/*v*, peptide concentration equal to 411 μM, pH 4.52 and 6.84 at the highest and lowest TFE percentages, respectively. NMR spectra for KIRCONG i/i+7 were recorded in H_2_O/TFE 60/40 *v*/*v* (peptide concentration 411 μM, pH 4.65). The volume of all NMR samples was equal to 500 μL. To conduct NMR structural analyses the following 2D [^1^H, ^1^H] experiments were recorded: TOCSY (Total Correlation Spectroscopy) [54], NOESY (Nuclear Overhauser Enhancement Spectroscopy) [55], ROESY (Rotating Frame Overhauser Enhancement Spectroscopy) [56] and DQFCOSY (Double Quantum-Filtered Correlated Spectroscopy) [57]. Typical acquisition parameters included 16–64 scans, 128–256 FIDs in t1, 1024 or 2048 data points in t2. Mixing times for TOCSY experiments were set to 70 ms whereas, mixing times equal to 200 ms and 300 ms were used to record NOESY spectra; ROESY experiments were acquired with a mixing time equal to 250 ms. Residual water signal was suppressed through excitation sculpting [58]. A standard strategy was followed to gain proton resonance assignments [59]. Trimethylsilyl-3-propionic acid sodium salt-D4 (TSP) (99% D, Armar Scientific, Döttingen, Switzerland) was used as internal standard for chemical shifts referencing. The Varian software VNMRJ 1.1D (Varian/Agilent Technologies, Milan, Italy) was implemented to process NMR spectra that were next analyzed with the program NEASY [60] included in CARA (Computer Aided Resonance Assignment) (http://www.nmr.ch/, accessed on 2 March 2022).

Chemical shift deviations from random coil values for Hα protons (CSD) for KIRCONG i/i+7 in H_2_O/TFE 60/40 *v*/*v* were evaluated with the method by Kjaergaard and collaborators [61]. Random-coil chemical shift reference values refer to T = 25 °C and pH 4.65 in H_2_O/TFE 60/40 *v*/*v* (https://www1.bio.ku.dk/english/research/bms/sbinlab/randomchemicalshifts1, accessed on 2 March 2022).

### 2.5. NMR Structure Calculations and Analysis

The NMR solution structure of KIRCONG i/i+7 in H_2_O/TFE 60/40 *v*/*v* was calculated with the software CYANA (version 2.1) [62]. CYANA library entries for the non-standard amino acids β-Alanine, (R)-N-2-(7′-octenyl) alanine and (S)-2-(4′-pentenyl) alanine were generated with the CLYB software [63]. To simulate the olefinic linker, the distance between CZ1 atom of (R)-N-2-(7′-octenyl) alanine and CE atom of (S)-2-(4′-pentenyl) alanine was imposed equal to 1.34 Å. Distance constraints (i.e., upper distance limits) were generated from manual integration of peaks in 2D [^1^H,^1^H] NOESY spectrum (300 ms mixing time); the GRIDSEARCH module of CYANA software [62] was implemented to obtain angular constraints. 100 random conformers were initially generated and in the end the 20 structures provided with the lowest CYANA target functions and better obeying to experimental constraints, were selected as representative NMR conformers [62,64]. The software MOLMOL [65] was used to additionally analyze the NMR peptide structure and to generate images.

### 2.6. Serum Stability

KIRCONG analogues (~1 mg/mL, 500 μM on average), were incubated with fetal bovine serum (FBS) at 25% (*w*/*v*) at 37 °C as previously described [39]. At the following time points: 0, 3, 17, 20, 23 and 42 h, 50 µL aliquots of the solutions were mixed with 50 µL of 15% trichloroacetic acid (TCA) to allow the precipitation of serum proteins, they were stored at −20 °C for at least 15 min. After centrifugation (13,000 rpm for 15 min) the supernatants were recovered. Samples were analyzed by RP-HPLC on a HPLC LC-4000 series (Jasco) equipped with UV detector using a C18-Kinetek column from Phenomenex (Milan, Italy), by employing a gradient from 5 to 70% of B (acetonitrile 0.1% TFA) versus A (water 0.1% TFA) in 20 min. Peptide compounds were detected by recording the absorbance at 210 nm and percentages were quantified by assuming 100% their peak areas at t = 0. All stability tests were performed at least in triplicates and reported data are averaged values.

### 2.7. Microscale Thermophoresis

MST experiments were carried out with a Monolith NT 115 system (NanoTemper Technologies, München, Germania) equipped with 40% LED and 40% IR-laser power. Labeling of His-tagged Catalytic Domain of JAK2 (residues 826–1132) (Carna Biosciences, Kobe, Japan) was achieved with the His-Tag labeling Kit RED-tris-NTA. The protein concentration was adjusted to 200 nM in labeling buffer (Nano Temper Technologies), while the dye concentration was set to 100 nM. Equal volumes (100 μL) of protein and fluorescent dye solutions were mixed and incubated at room temperature in the dark for 30 min. KIRCONG chim analogues were used in the following concentrations: KIRCONG amide 459 μM, KIRCONG disulfide 470 μM, KIRCONG i/i+5 525 μM, and KIRCONG i/i+7 585 μM in labeling buffer. Standard capillaries were employed for analysis, at 25 °C in 50 mM Tris-HCl, 150 mM NaCl, 0.05% Brij35, 1 mM DTT, 10% glycerol, 15% TFE buffer at pH 7.5, as already reported [51]. The equation implemented by the software MO-S002 MO Affinity Analysis [66], used for fitting data at different concentrations, is based on Langmuir binding isotherm.

## 3. Results

### 3.1. Design of Constrained KIRCONG Chim Mimetics

Our previous investigations on mimetics of SOCS3 pointed out that different protein regions provided hot spots for JAK2 recognition. In detail, KIRCONG chim peptide is a chimeric peptide mostly centered on KIR domain, since it includes the stretch 25–33 (while KIR spans 22–35 residues) covalently conjugated to a small aromatic stretch, named CONG (46–52 residues) (Table 1) that provides specific aromatic interactions with the catalytic domain of the kinase. This compound exhibited a low micromolar value of K_D_ through MST assay, but NMR studies performed in H_2_O revealed a flexible conformation lacking regular secondary structure elements. The presence of 15% TFE induced a slight decrease of flexibility of the compound that remained prevalently disordered, even if a certain helical content was encountered [50]. On this basis, herein we report the design and analysis of KIRCONG chim analogues in which structural modifications were inserted (Table 1) and their chemical structures are reported in Figure 1. These constraints were introduced with the aim to ameliorate drug-like features of mimetics indeed cyclization is a powerful approach to improve selectivity, metabolic stability, and bioavailability of bioactive peptides [39,67,68,69].

Different cyclic compounds were conceived: the first two are macrocycles containing disulfide and lactam bridges, respectively, between residues located in the C- and N-terminal extremities. In detail: KIRCONG amide was obtained through the formation of a peptide bond between the side chains of a Glu residue at the C-termini (native position 53 of CONG domain) and a non-native Lys upstream of KIR; in KIRCONG *disulfide* a cysteine bond was obtained between thiol groups of two non-native Cys (Table 1). After a detailed analysis of point mutations of SOCS3 residues that contact JAK2 protein, reported in [7], we introduced stapled constraints [70]. Bridges were generated by substituting residues not crucial for the interaction: in KIRCONG i/i+5, βAla and Ser^50^ and in KIRCONG i/i+7, Gln^32^ and Ser^50^ of reference KIRCONG chim (Table 1).

The α-helix is the most common secondary structure present in nature but different helical structures can be found in proteins, as 3_10_- and π-helix, even if less frequently [71]. To evaluate the effects of different helical structures into KIRCONG chim we introduced stapling residues at different positions: (i) i and i+7 to generate a α-helix and (ii) i and *i+5* to *f*orm of the so-called π-helix [71,72]. As consequence, in KIRCONG i/i+7 the bridge encompasses both KIR and CONG stretches, while in KIRCONG i/i+5 the cycle is located only in CONG region leaving the KIR region more flexible. 

To study the lipophilicity of designed compounds, we evaluated LogP values through the classical shake flask method [53] and reported them in Table 1. All KIRCONG analogues exhibited negative values except KIRCONG i/i+7, indicating its minor water solubility with respect to others.

### 3.2. Conformational Studies of Constrained KIRCONG Analogues

The conformational features of KIRCONG analogues were investigated through CD and NMR spectroscopies. 

#### 3.2.1. Circular Dichroism

CD spectra were recorded in the far UV region, for the low water solubility of all analogues, their stock solutions were prepared in 100% TFE and then diluted in aqueous buffer, till 15% *v*/*v* (TFE/aqueous buffer). The overlays of CD spectra are reported in Figure 2. As expected, for their non-native construction, KIRCONG analogues did not present canonical CD profiles. By comparing their spectral features with that of the lead compound [73] at the lowest TFE percentage (15%) only KIRCONG amide presents the slight positive band at ~235 nm due to an aromatic contribution to the peptide conformation. In this solvent system, also the deconvolution of CD spectra (Appendix A) indicates prevalent random + beta contents, particularly evident for the minimum at ~218 nm for KIRCONG disulfide and KIRCONG i/i+7 (Figure 2B,D). For both stapled compounds (Figure 2C,D) higher helical contents were already present at 15% TFE (8.2 and 3.8%, respectively) with respect to the other two analogues, for which helix percentage was 0 (Appendix A). Increasing amounts of TFE induced more ordered conformations especially for stapled structures: for them a clear transition toward helical conformations was detected, as testified by the progressive appearance of a secondary minimum at 220 nm.

#### 3.2.2. NMR Studies

Since stapled analogues are generally characterized by an increase of helical conformations with respect to linear counterparts, NMR studies coupled to CD analyses are useful to reveal such tendencies [74].

NMR characterization of stapled analogues was conducted under similar experimental conditions to those employed for KIRCONG chim [51]. Indeed, NMR spectra of KIRCONG i/i+5 were recorded in H_2_O/TFE mixtures containing increasing amounts of TFE (e.g., 15 and 40%) (Appendix A). By increasing TFE amount, NMR spectra showed a certain improvement of dispersion (Appendix A) likely indicating an increase of folded peptide population. However, a better comparison of 2D NMR experiments (Appendix A) showed a similar number of NOE peaks at both TFE concentrations highlighting only a minor improvement of structuration. By analyzing TOCSY and NOESY spectra many protons chemical shifts were identified although the assignments resulted ambiguous, particularly in the peptide region encompassing residues 10-16 (KIRCONG i/i+5, peptide numeration), where the non-natural residues (β-Alanine and (S)-N-2-(4′-pentenyl) alanine) are located (Appendix A). Within this region many duplicated spin systems are detected; this variability could be due to a CIS-TRANS equilibrium around the double bond of the staple coupled to the larger backbone flexibility induced by the β-Alanine. The presence of multiple conformers and the ambiguity of certain signals hampered to assign all proton residues and achieve a reliable structural model.

Better results were obtained with KIRCONG i/i+7, for which NMR spectra were recorded in solution at 40% TFE (Appendix A). Indeed, this compound resulted poorly soluble at NMR concentration in 15% TFE (Table 1): this is likely due to the presence of the longer alkyl chain of the i/i+7 stapled analogue. The comparison of TOCSY and NOESY spectra allowed to clearly assign almost completely the resonances of peptide protons (Appendix A) [59]. The presence of a strong contact between the Hζ1 and Hε protons in the spectral region around 5 ppm (Appendix A), indicated a trans arrangement of the double bond. The analysis of chemical shifts for Hα protons with respect to the random coil reference values showed a negative trend (Figure 3A), mainly, [75,76] that is characteristic of helical/turn conformations, more evident in the region between residues Y^7^ and V^16^, including the cyclic arrangement of the stapled peptide. The NOE pattern (Figure 3B) even if not canonically defined in helical conformation, presented a few signals of the type Hα-Hβ i/i+3 and Hα-H_N_ i/i+4 pointing out some helical content [59] in the peptide fragment 5-12. The NMR structure of KIRCONG i/i+7 (Figure 3C,D, Appendix A) is represented by a distorted helical arrangement extending through the whole sequence and appearing more regular in the C-terminal region between Y^12^ and (S)-N-2-(4′-pentenyl) alanine^15^ (KIRCONG i/i+7, peptide numeration) due to the presence of the hydrocarbon stapling (Figure 3D). Analysis of the NMR ensemble with the software MOLMOL [65] revealed the presence of a few backbone H-bonds characteristic of α-helix (i.e., (S)-N-2-(4′-pentenyl) alanine^15^ H_N_-F^11^ _C_O in 12/20 conformers; V^16^ H_N_-Y^12^ _C_O in 20/20 conformers; G^18^ H_N_- S^14^ _C_O in 5/20 conformers).

### 3.3. Serum Stability

To evaluate if introduced chemical modifications in KIRCONG analogues influence their stabilities to proteolytic degradation, pure compounds were incubated with fetal bovine serum (FBS) and the decrease of chromatographic peaks was followed during time In Figure 4, the area percentages of new derivatives of KIRCONG chim versus time are reported. After 3 h of incubation, KIRCONG chim is degraded by 30% while KIRCONG disulfide of 20% and others less than 10%. Interestingly, in both stapled peptides, the presence of unnatural amino acids used for staples formation, greatly increased their stability: after 42 h both peptides still showed a residual concentration of 85% and 75%, respectively while KIRCONG chim peptide appeared completely degraded.

### 3.4. MST Investigations

To evaluate the ability of KIRCONG analogues to recognize JAK2 catalytic domain, in vitro MST experiments were carried out (Figure 5), keeping JAK2 concentration constant and increasing ligands’ concentrations.

Even if all signals exhibit a dose–response curve for all four cyclic peptides (capillary shape and scan are reported in Appendix A), for KIRCONG i/i+5 and KIRCONG i/i+7 the signal variation was not meaningful and did not allow to obtain K_D_ values [77,78]. Concerning the other two analogues, KIRCONG amide and KIRCONG disulfide, the signal did not reach saturation and the fitting of data provided very high micromolar K_D_ values (Figure 5 and Table 1). 

## 4. Discussion

Given the key role that the JAK/STAT pathway plays in many disorders, the identification of molecules capable of modulating its activity, is a powerful strategy for the treatment of several serious diseases. In last years, many researchers have focused their attention on the identification of various small molecules capable to interfere with this pathway by inhibiting the activity of the JAK proteins and many of them have already been approved by U.S. Food and Drug Administration (FDA) (e.g., two JAK inhibitors (JAKins), Ruxolitinib for myeloproliferative disorders [79] and Afatinib in non-small-cell lung carcinoma (NSCLC) [80]). In this context, a parallel and more specific approach concerns the employment of mimetics of endogenous regulators of the JAK/STAT signaling, as SOCS proteins. Recently, we developed a mimetic of SOCS3, named KIRCONG chim, including two non-contiguous regions able to bind to JAK2 that also exhibited anti-inflammatory and antioxidant properties [51]. By employing KIRCONG chim as template compound, herein we present SAR investigations of several analogues bearing different cycles in the monomeric structure, obtained by the insertion and/or substitution of non-crucial residues of linear peptide counterpart. 

In detail, four cyclic analogues of KIRCONG chim were conceived: the first two, KIRCONG amide and disulfide, contained both a head-to-tail macrocycle where the cycle was obtained through the side chains of residues located at the opposite extremities of the sequence. In them the macrocyclization does not induce a significant limitation of flexibility (as observable from CD analysis) allowing however the recognition of JAK2 catalytic domain even if with lower affinity with respect to KIRCONG chim (Table 1). As expected, these kinds of cycles, more “natural” with respect to the others two, do not produce a significant improvement of stability to proteases after 24 h of analysis. 

On the other hand, in KIRCONG i/i+5 and i/i+7, the stapled linkages arise from side-chain-to side-chain non native bonds and are shorter with respect to *amide* and *disulfide* and involve different stretches of the chimeric structure: in the first analogue the cycle encompass only CONG region (its N-terminal residue is located in the β-alanines linker) while in the second both KIR and CONG fragments are involved in the staple.

From a conformational point of view, by comparing NMR structures of KIRCONG i/i+7 with KIRCONG chim [51] (both in H_2_O/TFE 60/40 *v*/*v*), the linear reference peptide exhibited a more ordered helical conformation (Appendix A). This result indicates that the co-existence of flexible β-alanines in the linking portion and of the hydrocarbon staple does not generate a rigid helical arrangement, generally typical of most stapled peptides with i/i+7 pattern [81]. On the other hand, the *i/i+5* arrangement in KIRCONG fails to generate a well folded structure and the compound assumes multiple conformational states. 

The structural stiffening of KIRCONG chim in both stapled cases, provided a negative effect on the ability of the SOCS3 mimetic to recognize JAK2: MST assay, in the investigated concentration range (0.015–500 μM on average), did not provide a significant dose-response variation of the signal and the very limited water solubility hampered to increase ligands’ amounts. In addition, the same binding assay employing the linear version of KIRCONG i/i+5 peptide, reported in Appendix A, provided a good dose-response of the MST signal and a K_D_ value (8 µM) that is quite similar to that of KIRCONG chim (11 µM) (Table 1). This experiment confirmed that the absence of cyclization restores the ability to bind JAK2. It is important to point out that the presence of stapled cycles provided greater stabilities to proteases till 42 h. 

In conclusion, overall data indicate, how in the cyclization of bioactive peptides, a crucial role is exerted by the balance between limitation of flexibility of the backbone and right exposure of hot spots of interaction. The clear modulating effect by the length and the position of the cycle within the chimeric structure reported by this study, could help, in the future, the design of more soluble mimetics of SOCS3 as anti-inflammatory agents.

## Figures and Tables

**Figure 1 pharmaceuticals-15-00458-f001:**
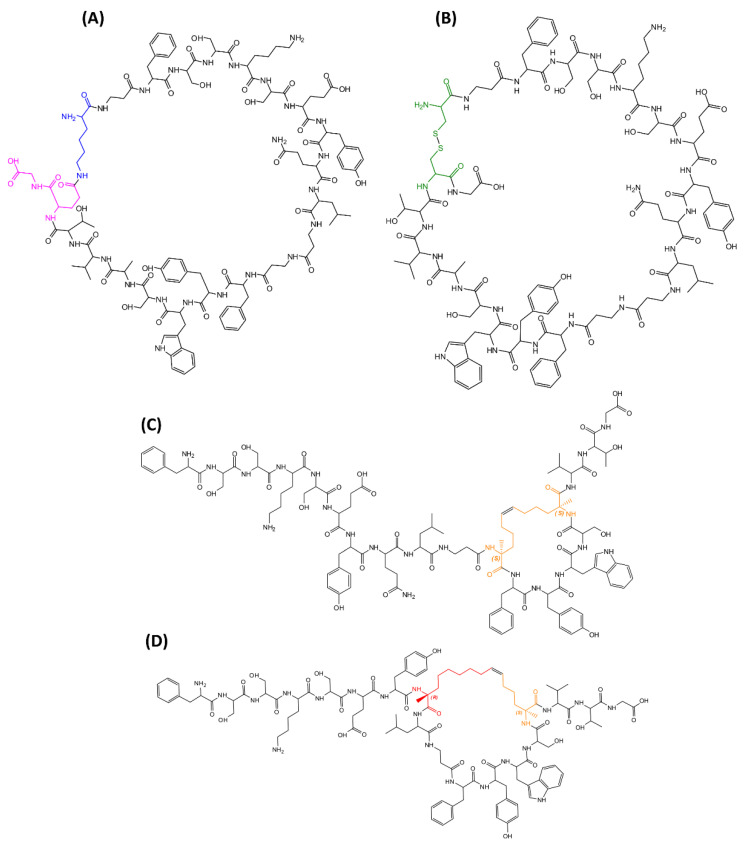
Chemical structures of KIRCONG analogues with residues involved in cycle formation colored: (**A**) KIRCONG amide (blue: Lys, purple: Asp), (**B**) KIRCONG disulfide (green: Cys), (**C**) KIRCONG i/i+5 (orange: (S)-N- 2-(4′-pentenyl) alanine) and (**D**) KIRCONG i/i+7 (orange: (S)-N-2-(4′-pentenyl) alanine, red: (R)-2-(7′-octenyl) alanine).

**Figure 2 pharmaceuticals-15-00458-f002:**
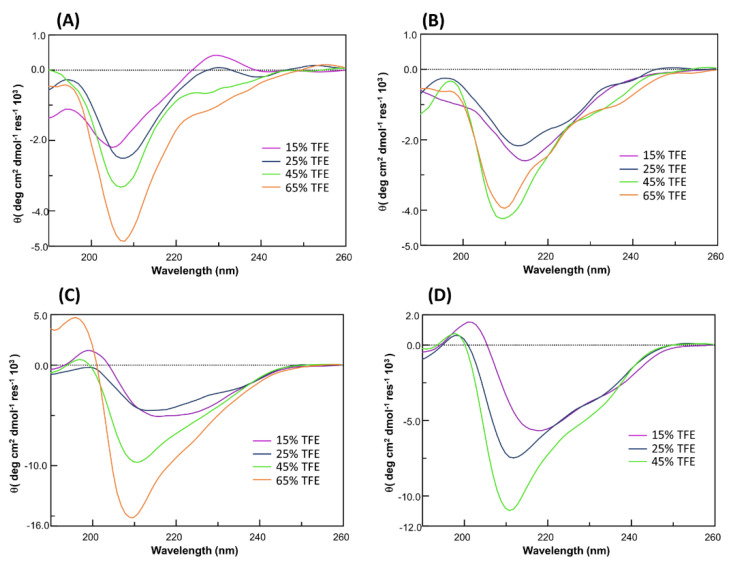
CD spectra of: (**A**) KIRCONG amide; (**B**) KIRCONG disulfide; (**C**) KIRCONG i/i+5; (**D**) KIRCONG i/i+7 in TFE/buffer 15-65% *v*/*v*. KIRCONG i/i+7 resulted not soluble in 65/25 *v*/*v*, TFE/buffer and the related spectrum is absent.

**Figure 3 pharmaceuticals-15-00458-f003:**
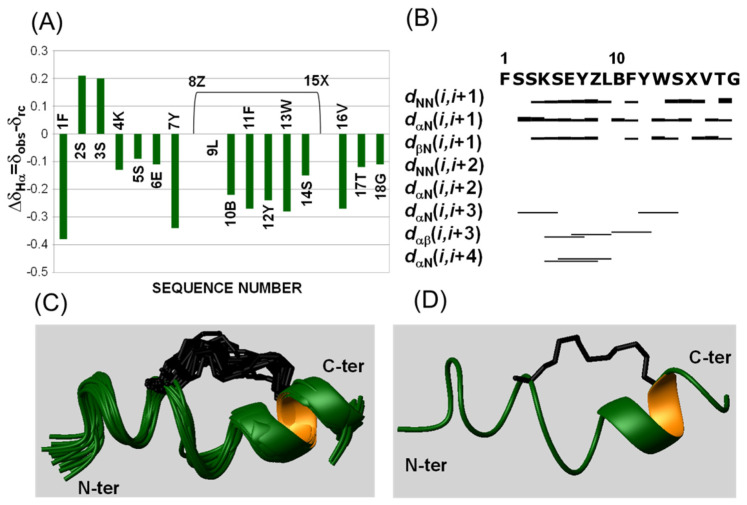
NMR analysis of KIRCONG i/i+7 in H_2_O/TFE 60/40 *v*/*v*. (**A**) Chemical shift deviations of Hα protons from random coil values (∆δ_Hα_). Standard amino acids are reported with the one letter code whereas, B stands for β-Alanine, Z stands for (R)-N-2-(7′-octenyl) alanine and X stands for (S)-2-(4′-pentenyl) alanine. ∆δ_Hα_ for Z and X is set equal to 0 as the reference random coil value is missing. For ∆δ_Hα_ evaluation β-Alanine was assimilated to Glycine. (**B**) NOEs pattern. (**C**) Ribbon representation of KIRCONG *i/i+7* NMR structures: 20 conformers are superimposed on the backbone atoms of residues 3–17, and (**D**) ribbon representation of the first NMR conformer. The aliphatic linker between (R)-N-2-(7′-octenyl) alanine (residue n.8) and (S)-2-(4′-pentenyl) alanine (residue n.15) is shown in black. The NMR structure was generated from 261 upper distance limits (170 intraresidue, 58 short-range, 27 medium-range and 6 long-range) and 73 angular constraints.

**Figure 4 pharmaceuticals-15-00458-f004:**
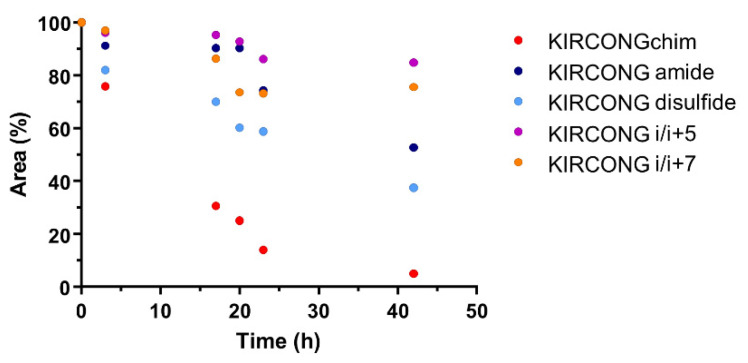
Serum stability assay of KIRCONG chim derivatives. The serum stability was evaluated by incubation in 25% FBS for 42 h. Residual peptide amount is expressed as the percentage of the initial amount versus time.

**Figure 5 pharmaceuticals-15-00458-f005:**
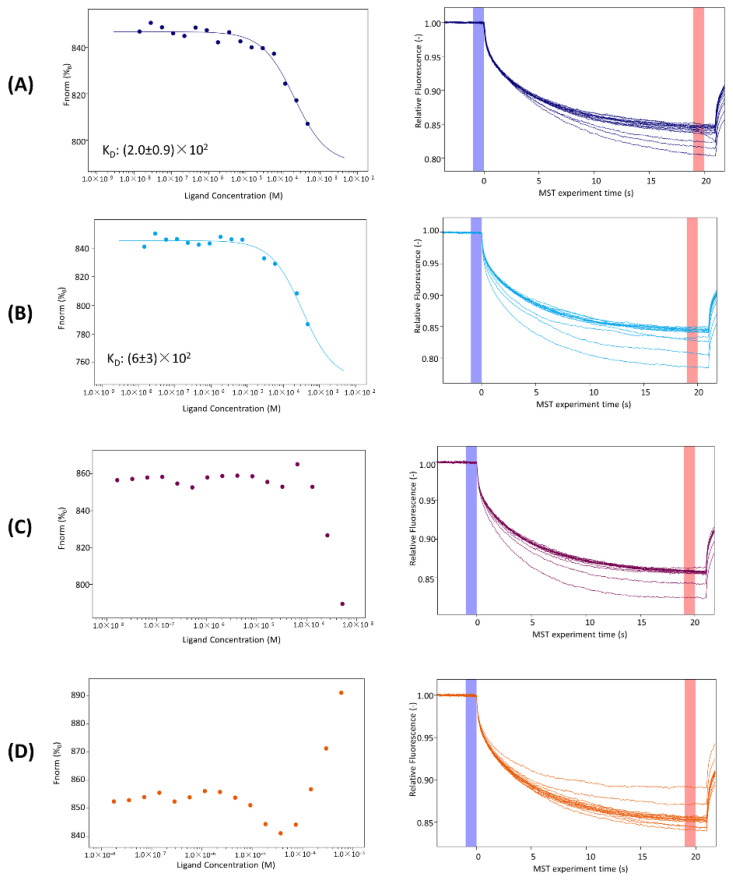
Left column: binding isotherms for MST signals versus peptide concentrations. Right column: thermophoretic traces of MST assays for the binding to JAK2 of (**A**) KIRCONG amide; (**B**) KIRCONG disulfide; (**C**) KIRCONG i/i+5; (**D**) KIRCONG i/i+7.

**Table 1 pharmaceuticals-15-00458-t001:** Sequences and names of compounds investigated in this study. Residues belonging to different human SOCS3 regions are colored in blue (KIR) and orange (CONG). X: (S)-2-(4′-pentenyl) alanine Z: (R)-2-(7′-octenyl) alanine).

Name	Sequence	K_D_ (µM)	LogP
KIRCONGchim [51]	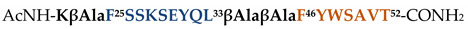	(1.1 ± 0.3) × 10	−1.32
KIRCONG amide	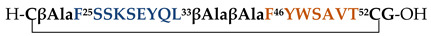	(2.0 ± 0.9) × 10^2^	−0.92
KIRCONG disulfide	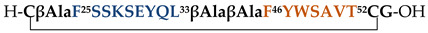	(6 ± 3) × 10^2^	−1.46
KIRCONG i/i+5	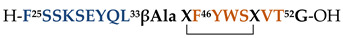	No binding	−0.27
KIRCONG i/i+7	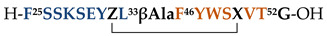	No binding	0.29

## Data Availability

The data is contained within the article and Appendix A.

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
