# Peer review of "Structure-Activity Relationship Investigations of Novel Constrained Chimeric Peptidomimetics of SOCS3 Protein Targeting JAK2"

_pharmaceuticals, 2022, doi:10.3390/ph15040458_

Round 1

Reviewer 1 Report

With the aim to improve drug-like features of KIRCONG, herein the authors report novel cyclic analogues bearing different linkages. In detail, in two of them hydrocarbon cycles of different lengths were inserted at positions i/i+5 and i/i+7 to improve helical conformations of mimetics. Structural features of cyclic compounds were investigated by CD (Circular Dichroism) and NMR (Nuclear Magnetic Resonance) spectroscopies while their ability to bind to catalytic domain of JAK2 was assessed through MST (MicroScale Thermophoresis) assay as well as their stability in biological serum. Overall data indicate a crucial role exerted by the length and the position of the cycle within the chimeric structure and could pave the way to the miniaturization of SOCS3 protein for therapeutic aims. 

Very interesting work of notable scientific value that can open new avenues of research in the field of peptidomimetics. In my opinion, it is worth publishing with some small corrections in the English.

Author Response

R1.1: With the aim to improve drug-like features of KIRCONG, herein the authors report novel cyclic analogues bearing different linkages. In detail, in two of them hydrocarbon cycles of different lengths were inserted at positions i/i+5 and i/i+7 to improve helical conformations of mimetics. Structural features of cyclic compounds were investigated by CD (Circular Dichroism) and NMR (Nuclear Magnetic Resonance) spectroscopies while their ability to bind to catalytic domain of JAK2 was assessed through MST (MicroScale Thermophoresis) assay as well as their stability in biological serum. Overall data indicate a crucial role exerted by the length and the position of the cycle within the chimeric structure and could pave the way to the miniaturization of SOCS3 protein for therapeutic aims.  Very interesting work of notable scientific value that can open new avenues of research in the field of peptidomimetics. In my opinion, it is worth publishing with some small corrections in the English.

A1.1 We thank the reviewer for his/her favorable comments. We have revised the English of the manuscript and variations are highlighted in red

Reviewer 2 Report

The objective of this study was to develop analogues of a chimeric peptide named KIRCONG to improve its drug-like features, including the lipophilicity and binding affinity for JAK2. 

Major points

Line 138 - 139: "... we calculate LogP values through Molinspiration server ...". The authors should carry out a simple experiment to determine octanol-water partition coefficient values and then LogP values to verify if the model predicted LogP values match the experimentally determined LogP values.

Although microscal thermophoresis assay can be used to evaluate the binding of compound to the kinase protein, it does not show to what extent the binding would inhibit the kinase activity. In this regard, besides the microscale thermophoresis assay, the authors should consider conducting the protein tyrosine kinase assay to determine the IC50 values of KIRCONG and its individual analogues and then compare the inhibitory effect on JAK2 activity among different compounds.  

Author Response

Reviewer 2

R2.1 The objective of this study was to develop analogues of a chimeric peptide named KIRCONG to improve its drug-like features, including the lipophilicity and binding affinity for JAK2. 

Major points

Line 138 - 139: "... we calculate LogP values through Molinspiration server ...". The authors should carry out a simple experiment to determine octanol-water partition coefficient values and then LogP values to verify if the model predicted LogP values match the experimentally determined LogP values.

A2.1 We thank the reviewer for his/her comment and to allow us to improve the scientific quality of our study.

Accordingly, we performed experiments to determine LogP values following the classical shake flask method [1]. Experimental data provided LogP values quite different from those obtained through the Molinspiration server (Table R1), probably because the server is optimized for small molecules and branched structures and not for macrocyclic peptides. Therefore, we consider more reliable experimental LogP values and substitute theoretical data with experimental ones. Accordingly, we delete references to molinspiration while added a new paragraph in material and method section and results were appropriately included and discussed in the revised version of the manuscript.

Table R1. Comparison of theoretical and experimental LogP values.

Name

LogP (theoretical)

LogP (experimental)

KIRCONGchim

-5.99

-1.32

KIRCONG amide

-6.27

-0.92

KIRCONG disulfide

-6.23

-1.46

KIRCONG i/i+5

-5.52

-0.27

KIRCONG i/i+7

-4.35

0.29

R2.2: Although microscal thermophoresis assay can be used to evaluate the binding of compound to the kinase protein, it does not show to what extent the binding would inhibit the kinase activity. In this regard, besides the microscale thermophoresis assay, the authors should consider conducting the protein tyrosine kinase assay to determine the IC50 values of KIRCONG and its individual analogues and then compare the inhibitory effect on JAK2 activity among different compounds.  

A2.2 We completely agree with this observation, but we have to underline that we are not a molecular biology lab, and we buy a His-tagged recombinant catalytic domain of JAK2 and we usually employ ~0,7µg of protein for each MST assay. While in the case of tyrosine kinase assay, we should buy a i) fluorescence-based assay kit including additional amounts of JAK2 (entire or catalytic domain) with suitable fluorescent substrates and/or ii) recombinant JAK2 protein, usually greater amounts for mass-based analysis and also in this case appropriate substrates. We want also to highlight that KIRCONG, that is the starting compound, already demonstrated a direct inhibition of JAK2, just as the whole SOCS3, in cellular context, and a reduction of STAT3 phosphorylation as reported in [2]. Since in the KIRCONG analogues investigated in this study we did not change hot spots of interaction with JAK2, we can quietly assess that their mechanism of action is the same and occurs through the inhibition of catalytic activity of the kinase.

  1. OECD. Guidelines for the Testing of Chemicals, Test No. 107: Partition Coefficient (n‐octanol/water): Shake Flask Method.
  2. La Manna, S.; Lopez-Sanz, L.; Mercurio, F.A.; Fortuna, S.; Leone, M.; Gomez-Guerrero, C.; Marasco, D. Chimeric Peptidomimetics of SOCS 3 Able to Interact with JAK2 as Anti-inflammatory Compounds. ACS Med Chem Lett 2020, 11, 615-623, doi:10.1021/acsmedchemlett.9b00664.

Round 2

Reviewer 1 Report

The article in its initial form seemed to me worthy of being published. Now, with the small changes indicated by the referees, it has gained in quality and therefore I consider that it can be accepted for publication.

Reviewer 2 Report

The authors have addressed all my concerns.